# Learning Three Dimensional Tennis Shots Using Graph Convolutional Networks

**DOI:** 10.3390/s20216094

**Published:** 2020-10-27

**Authors:** Maria Skublewska-Paszkowska, Pawel Powroznik, Edyta Lukasik

**Affiliations:** Department of Computer Science, Lublin University of Technology, 20-618 Lublin, Poland; p.powroznik@pollub.pl (P.P.); e.lukasik@pollub.pl (E.L.)

**Keywords:** tennis movement recognition, ST-GCN, fuzzy data

## Abstract

Human movement analysis is very often applied to sport, which has seen great achievements in assessing an athlete’s progress, giving further training tips and in movement recognition. In tennis, there are two basic shots: forehand and backhand, which are performed during all matches and training sessions. Recognition of these movements is important in the quantitative analysis of a tennis game. In this paper, the authors propose using Spatial-Temporal Graph Neural Networks (ST-GCN) to challenge the above task. Recognition of the shots is performed on the basis of images obtained from 3D tennis movements (forehands and backhands) recorded by the Vicon motion capture system (Oxford Metrics Ltd, Oxford, UK), where both the player and the racket were recorded. Two methods of putting data into the ST-GCN network were compared: with and without fuzzying of data. The obtained results confirm that the use of fuzzy input graphs for ST-GCNs is a better tool for recognition of forehand and backhand tennis shots relative to graphs without fuzzy input.

## 1. Introduction

Human action recognition is a dynamically developing field of computer vision. It has reached a great interest in sport analysis, especially in video analysis. This technology has gained a great popularity in obtaining statistics of sports, sports techniques analysis, and understanding sports tactics [1].

Optical motion capture systems are a very popular method of precisely recording an athlete’s movements. Based on markers attached directly to the skin or to a special suit, it is possible to capture both the participant’s body and additional objects such as the tennis racket. The obtained information is often used to verify movements and the athlete’s progress, and develop a new training method or adjust the current one to the newest requirements. Changing the three-dimensional position of the markers allows to observe even the smallest movements with high accuracy. Additional biomechanical models allow the calculation of additional analogous parameters such as angles and moments. The whole set gives a sophisticated tool for sport analysis.

Machine learning using convolutional neural networks is currently undergoing great progress in action recognition [2,3,4]. Moreover, images are often processed using graphs, where a given pixel denotes the node of the graph, connected to the adjacent pixels. This approach has achieved a great success especially in human motion recognition, because the joints in the image correspond to the human topology. That is why in the research GNNs are connected with convolutional networks (GCNs). Spatial-Temporal Graph Neural Networks (ST-GNNs) are often used in image and video processing, especially for identifying human action patterns [5,6,7,8,9], but also for image classification [10] or semi-supervised learning [11]. This method is of great interest, because it is able to perform automatic analysis based on spatial configuration and by temporal dynamics.

In tennis there are two basic shots: forehand and backhand, which are performed during all matches and training. Recognition of these movements is important in the quantitative analysis of tennis players. In this paper the authors combine a sophisticated method of acquisition together with convolutional neural networks. The new method is proposed for recognising of two basic tennis movements. The ST-GCN network was selected to challenge the above task. It was also to verify how data fuzzyfication affects final recognition. Recognition of the shots is performed on the basis of images obtained from the 3D tennis movements recorded by Vicon motion capture system (Oxford Metrics Ltd, Oxford, UK). The images contain the participant of the study together with the tennis racket. The main aim of this paper is to prove the thesis that “the use of fuzzy input graphs for ST-GCN improves the recognition of forehand and backhand tennis shots relative to graphs without fuzzy input”.

Deep neural networks are very widely used in recognising human movements. Convolutional Neural Networks (CNNs) play an important role in the study of skeleton-based action recognition [12,13]. This task employs Recurrent Neural Networks (RNNs). The end-to-end hierarchical RNN is proposed for skeleton-based action recognition in [14]. A fully connected deep Long Short-Term Memory (LSTM) network is introduced in [15]. This is used to learn the co-occurrence features of skeleton joints. A two-stream RNN architecture to model both temporal dynamics and spatial configurations for skeleton-based action recognition is proposed in [16]. Two different structures for the temporal stream: a stacked RNN and a hierarchical RNN, designed according to human body kinematics, are explored. An ensemble temporal sliding LSTM (TS-LSTM) network for skeleton-based action recognition is used to capture short-, medium- and long-term temporal dependencies, and even spatial skeleton pose dependency [17]. The study of a model with spatial reasoning and temporal stack learning (SR-TSL) based on the SYSU 3D Human-Object Interaction dataset is described in [18]. It was in 1997 that a neural network was first combined with graphs [19]. There are four types of graph neural networks: recurrent graph neural network, convolutional graph neural network, graph autoencoders, and spatial-temporal graph neural network [5].

CNNs have recently been used in the spectral domain relying on the Laplacian graph [20,21]. In [10,22,23] the authors apply the convolution directly on the graph nodes and their neighbours, which construct the graph filters on the spatial domain. The graph convolutional neural networks are applied to skeleton-based action recognition in [6]. The model of dynamic skeleton Spatial-Temporal Graph Convolutional Networks (ST-GCNs) is formulated on top of a sequence of skeleton graphs, where each node corresponds to a joint of the human body and there are two types of edges: the spatial ones, which conform to the natural connectivity of joints and the temporal ones, which connect the same joints across consecutive time steps. The authors propose several principles in designing convolution kernels in an ST-GCN in order to use it for skeleton modelling [6]. ST-GCNs automatically learn both the temporal and spatial features from the skeleton data and achieve remarkable performance in skeleton-based action recognition. An ST-GCN just learns local information on a certain neighbourhood, but does not capture the correlation information between all the joints (i.e., global information). A model of dynamic skeletons called attention module-based-ST-GCN introduces global information. The attention module brings stronger expressive power and generalisation capability [24].

Li et. al. [8] propose a new model for the hand gesture graph convolutional network (HG-GSN) based on the previous work. The skeleton-based action recognition based on a two-stream adaptive graph convolutional network is described in [25]. A novel graph convolutional networks-hidden conditional random field (GCN-HCRF) model is presented in [9]. It retains the spatial structure of human joints from the beginning to end.

Another application of GCNs on images is visual question answering that explores the answers to the questions about images [26]. GCNs are exploited to understand the visual relationships among the objects which are on the images, which helps to characterise the interactions among them [27]. An architecture of GCN and LSTM are proposed to explore the visual relationships for image capturing [28].

A few papers about tennis movement analysis have been prepared recently. Their results may be used for the tactic analysis or for the recognition of the tennis movements. Most of the research is based on the broadcast videos or videos obtained from THETIS dataset [29]. Three tennis recognition movements: serve, hit and non-hit are presented in [30]. The authors described an application of transductive transfer learning for video annotation. Sudden changes of the tennis ball were detected. Presentation of two basic shots (forehand and backhand) from a broadcast tennis video is found in [31]. The data obtained were used in the multimodal framework which is for tactics analysis. A classification of twelve tennis movements based on the human action dataset is presented in [29]. The set consists of video clips captured by Kinect, gathered in the THETIS dataset. These data may provide the depth map of motion data, but also a 3D skeleton scheme. The 12 non-linear SVM classifier was used for verifying the accuracy of the described method [32]. Based on the RGB videos from the THETIS dataset fine-grained action recognition in tennis was performed using deep neural networks. The authors presented a model for classifying the videos into 12 actions from raw footage. The algorithm used the Inception convolutional neural network for extracting features which are further processed by the 3-layered LSTM network. The method reached 43.2% of accuracy. Another research based on the THETIS dataset concerning also recognition of 12 tennis movements is presented in [33]. Two classifiers were used for this purpose: an SVM and a linear-chain CRF. The accuracy was reached at the level of 86%. The recognition of 12 tennis movements based on two datasets—THETIS and HMDB51—is presented in [1]. The authors used convolutional neural network for extracting spatial representations and a multi-layerd LSTM network for historical information. The maximum accuracy was reached at the level of 95%. An action recognition of forehand and backhand by a tennis player in broadcast video is proposed in [34]. The discussed method used histograms based on optical flow. Recognition of the actions is performed by the SVM classifier.

According to the authors’ knowledge, there is no research considering the recognition of tennis movements using graphs together with images consisting of models of both the human body and a tennis racket. Due to the fact that the recognition of tennis movement is an important aspect of sport analysis, in this paper research is performed using ST-GCNs to recognise two tennis strokes: forehand and backhand. It was also verified if the accuracy obtained from the literature can be improved.

## 2. Material and Methods

### 2.1. Capturing Motion Data

Ten male tennis coaches (aged 23.85±5.82, height 1.80±0.11 m, weight 73.21±8.98 kg) took part in the research. Nine were right-handed and one was left-handed. The length of the internship as a coach was the decisive element. They signed the consent for the study. A passive optical motion capture system was used to track the participant and the racket while performing tennis strokes at the Laboratory of Motion Analysis and Interface Ergonomics at the Lublin University of Technology.

The participant was prepared for the experiment according to the Plug-in Gait Model. In total, 39 retroreflective markers were attached to the participant using hypoallergenic double-sided tape as specified in the model. This model allows for calculating angles, torques and forces in a subject’s joints. Seven retroreflective markers were also attached to the tennis racket. One marker was attached to the top of the racket head, two on both sides of the racket, one in the bottom of the racket head and one to the bottom of the racket handle. They allow both to reconstruct the shape of the racket and further analyse the racket’s movement.

Each participant was measured for the purpose of creating and scaling a new subject in the Vicon Nexus software. The dimensions measured were: height, weight, leg length, arm offset, knee, ankle, elbow and the thickness of both hands. The subject’s calibration was performed as the next step in preparation. Each participant performed two separate movements: forehand and backhand, while running and avoiding a bollard placed on the floor. Because the participant was running, the strokes were more natural than hitting the ball from a standing position. At first, ten forehand strokes without a ball were performed, followed by ten backhand strokes without a ball. Next, these exercises were repeated with a ball. The participant hit a ball which was caught by a special net.

Each 3D recording was post-processed using the Vicon Nexus software. The process consisted of four main steps: marker labelling, gap filling using interpolation methods, data cleaning (e.g., deleting all unlabelled markers) and applying the Plug-in-Gait model (only for human body). A new subject was created for the racket. It consisted of seven markers. The post-processed recordings were exported as C3D files.

### 2.2. Spatial Temporal Graph

From the post-processed C3D files the films were created using the Vicon Nexus 2.0 software. The images were obtained from the recordings by the ffmpeg tool with the sampling equal to 0.1 s. They were divided into three categories: forehand shot, backhand shot and no shot. The last type of the images present the tennis player with the racket by performing running between the shots. Both forehand and backhand shots were further divided into two phases: before shot (the preparation phase) and direct shot. The preparation phase was indicated from the moment of starting performing the racket swing untill before the moment of hitting the ball. The second phase was indicated as the movement of impact and the racket swing after the shot. This attitude gives the possibility of creating a temporal graph.

Skeleton data can be obtained from motion capture devices as well as using algorithms to evaluate motion in the image. Usually the data are a sequence of frames and the way they are connected is represented by a set of coordinates. Considering the sequence of the position of joints in 2D or 3D coordinates, changing in time, a spatial time graph with joints as graph nodes and the edges of the graphs corresponding to the time representation was obtained. In the research carried out, some set of the points obtained during the measurements was redundant and thus replaced, e.g., a set of points representing a tennis racket was replaced by two graph nodes and an edge between them. Thus, the structure of the input data is expressed in the form of a graph G=(V,E) and consists of *N* nodes corresponding to *N* joints and changes in their position in time. The node set $V=\{v_{ti}|t=1,\ldots ,T,i=1,\ldots ,N\}$ describe all joints in a skeleton. The example of input data is presented in Figure 1.

Before proceeding to the proper classification all recorded data were transformed as follows. The black area appearing in each frame (Figure 2) has been replaced by white. Further transformations were carried out on the skeleton isolated from the white background. The excess elements of the skeleton were replaced by a smaller number of points, i.e., the racket was replaced by two points (one reflecting the handle, the other the head of the racket) and the edge connecting them. Similarly, the points representing the human head were replaced by one point with interpolated coordinates. The above operations did not affect the quality of the classification, but provided faster calculation and reduced the number of entry points.

### 2.3. Recognition of Tennis Shots

Due to the unchanging location of joints and the human skeleton, it is possible to recognise movements using motion capture technology. The use of appropriate algorithms means that this technology is also resistant to changes in the lighting or scenery in which tests are performed [35]. There is therefore a wide spectrum of methods by which it is possible to identify human movements. Existing approaches can generally be divided into methods that use previously identified input parameters and methods based on deep learning techniques [36]. During specific movements, there are changes within direct joints and small body fragments connected with it. Existing motion recognition methods have confirmed the effectiveness of introducing body parts for modelling [37,38]. Due to the representation of changes in specific areas of the body over time, a hierarchical structure that is the basis for classification is obtained. The input to the neural network is therefore the joint coordinate vectors on the graph nodes. Multiple layers of spatial-temporal graph convolution operations will be applied on the input data, as a result of which further network layers will be generated. The final classification will be done using the standard Sotfmax classifier, and the model will be learned using stochastic gradient descent algorithm. The basis of the ST-GCN network is constituted by revolutionary neural networks. For a single image, the convolution operation allows to process two-dimensional input data into a two-dimensional set of features. Given adequate data padding, the size of the input and output data can be the same size. Thus, for a K×K-size kernel, the convolution operation allowing the projection of input features $f_{in}$ in output value $f_{out}$ at the spatial location *x* can be written as [6]:(1)$$\begin{array}{cc} f_{out} & =\sum^{K}\sum^{K}f_{in}p\left(x\right)*w \end{array}$$
where *p* is the sampling function $Z^{2}$x$Z^{2}\to Z^{2}$ and *w* is the weights function $Z^{2}\to R^{2}$. It should be emphasised that the weight function is irrelevant to the location of the *x* point. Standard convolution is therefore achieved by encoding the rectangular grid in p(x). Detailed explanations can be found in [39]. The sampling function can be defined on the neighbour set $G\left(v_{ti}\right)=\left\{v_{tj}|d\left(v_{tj},v_{ti}\right)\leq D\right\}$ of node $v_{ti}$, where $d(v_{tj},v_{ti})$ indicates the minimum length of any path from $v_{tj}$ to $v_{ti}$. In this research, *D* equals 1. Considering the above it can be written that [6]:(2)$$\begin{array}{c} p:G(v_{ti})\to V. \end{array}$$
Therefore, for graphs, dependence (1) can be presented as follows [6]:(3)$$\begin{array}{cc} f_{out} & =\sum_{v_{tj}\in G\left(v_{ti}\right)}f_{in}\left(v_{tj}\right)*w\left(v_{tj}\right). \end{array}$$

For the implementation of the ST-GCN network, the algorithm proposed by [11] was used. For a single motion capture frame, the connection diagram between joints has been described using matrix *A*, while matrix *I* describes connections between joints in subsequent frames. The above relationship, for a single frame, can be described in the following equation [6]:(4)$$\begin{array}{c} f_{out}=\Lambda^{-\frac{1}{2}}\left(A+I\right)\Lambda^{-\frac{1}{2}}f_{in}W, \end{array}$$
where $\Lambda^{ii}=\sum_{j}\left(A^{ij}+I^{ij}\right)$ and *W* corresponds to the weight vector. In practice, the input feature map is represented by a tensor of (C,V,T) dimensions. *V* represents the number of nodes (19 for each person in the tested database), *C* is the dimension of each node (2 in the conducted research) and the variable *T* is the number of time steps (14 in our dataset). The graph convolution is implemented as a standard two-dimention convolution, multiples with normalised adjacency matrix on the second dimension.

The classifier model consists of a 3-layer ST-GCN: pooling, convolution and volutional layer. The whole structure is presented in Figure 3. The first spatial temporal convolutional layer consists of 32 kernels, the second and third have 64 kernels each. The output from the third layer is the average pooled in joints and temporal directions and passed through a 1×1 convolutional layer. The output of this layer is connected with a layer of dimension 3 (corresponding to 3 identified states), followed by a softmax function to generate class labels. ReLU nonlinearity are followed after all ST-GCN layers and BatchNorm layer is also followed after all layers, omitting fully connected layer which was not shown in Figure 3 due to the clarity of the image. The active features box provide an important data for action recognition. It consists of two types of parameters: movement parameters, describing acceleration and velocity of single joints and posture data, characterizing distance and angles between joints in two triangular areas: lower (left feet, spine, right feet) and upper (left hand, head, right hand) limb. The data combined with ST-GCN output are passed through a fully connected layer.

**Definition** **1.**
*A non-empty fuzzy set $f_{s}$ can be understood as an ordered pair $\left(f_{s},\eta_{f_{s}}\right)$, where $\eta_{f_{s}}$ is a membership function $\eta_{f_{s}}:f_{s}\mapsto \left[0,1\right]$, which allows to perform the fuzzyfication operation. $\eta_{f_{s}}$ assigns to each element x in $f_{s}$ a degree of membership, 0≤σ(x)≤1 [40].*


**Definition** **2.**
*A fuzzy relation on $f_{s}$ is a fuzzy subset of $f_{s}$x$f_{s}$. A fuzzy relation $\eta_{f_{s}}$ on $f_{s}$ is a fuzzy relation on the fuzzy subset σ, if $\eta_{f_{s}}\left(x,y\right)\leq \sigma \left(x\right)\wedge \sigma \left(y\right)$ for all x,y from $f_{s}$ and ∧ stands for minimum. A fuzzy relation $\eta_{f_{s}}$ on $f_{s}$ is said to be symmetric if $\eta_{f_{s}}\left(x,y\right)=\eta_{f_{s}}\left(y,x\right)$ for all $x,y\in f_{s}$ [40].*


**Definition** **3.**
*A fuzzy graph is a pair $G:(\sigma ,\eta_{f_{s}})$ where σ is a fuzzy subset of $f_{s}$, $\eta_{f_{s}}$ is a symmetric fuzzy relation on σ [40].*


**Definition** **4.**
*$\left(\sigma^{\prime },\eta_{f_{s}}^{\prime }\right)$ is a fuzzy sub graph of $\left(\sigma ,\eta_{f_{s}}\right)$ if $\sigma^{\prime }\subseteq \sigma$ and $\eta_{f_{s}}^{\prime }\subseteq \eta_{f_{s}})$ [40].*


The use of fuzzy logic in neural networks allows modelling “uncertain” phenomena [41]. Therefore, the fuzzy graph neural network is able to imitate the way people perceive the environment. Applying fuzzy logic rules, sharp boundaries between the analysed sets are blurred. Fuzzification and defuzzification procedures allow to transform sets from one state to another. The input data fuzzification process was carried out in the Matlab environment using the following membership functions:(5)$$\begin{array}{c} \eta_{Rf_{s}}\left(x\right)=\left\{\begin{array}{c} 0,(x>d)\\ \frac{d-x}{d-c},c\leq x\leq d\\ 1,x<c \end{array}\right. \end{array}$$
(6)$$\begin{array}{c} \eta_{f_{s}}\left(x\right)=\left\{\begin{array}{c} 0,(x<a)||(x>d)\\ \frac{x-a}{b-a},a\;\leq x\leq b\\ 1,b<x<c\\ \frac{d-x}{d-c},c\leq x\leq d \end{array}\right. \end{array}$$
(7)$$\begin{array}{c} \eta_{Lf_{s}}\left(x\right)=\left\{\begin{array}{c} 0,(x<a)\\ \frac{x-a}{b-a},a\;\leq x\leq b\\ 1,x>b \end{array}\right. \end{array}$$
where a,b,c,d are trapezoidal function parameters and a<b<c<d. In the case of Equation (5) a=b=−∞, and in (7)c=d=∞.

## 3. Experiments and Results

The set of all registered poses was divided into seven subsets: a set of sequences containing preparation for forehand and backhand shots (separately), proper forehand and backhand shots, and sets containing poses immediately following after forehand and backhand shots. The last set contained poses unrelated to any of the above strokes.

The whole data set consists of 1080 images: 348 forehand shots (divided into three subsets), 354 backhand shots (divided into three subsets) and 378 poses without shots. An example of raw data is presented in Figure 2.

The graph constituting the input data was a component of three items: preparation for shot, shot, movement immediately after shot. The selection of elements from each subset took place randomly, within a given type of shot. Forehand and backhand stroke phases were not allowed to be mixed. For data not related to any of the strokes, all elements were from the last subset.

For each of the proposed classifiers, a series of tests were carried out consisting in randomly dividing the available data into a training set and a test set. When dividing the considered sets, different proportions were used between the sets, starting from 10% to 65% share of teaching collections. Three independent replicate trials were performed for each division and the results obtained were collected.

The effectiveness of ST-GCN classifiers (Figure 4) depends to a large extent on the selection of elements for the training set. Unfortunately, no correlation between the number of elements in the training set and the classification efficiency can be noted.

All results were obtained in the Matlab environment with the Parallel Computing Toolbox. The hardware platform was equipped with the Intel Core i7-9700KF 8x 3.60GHz processor and the NVIDIA GeForce GTX 1070 Ti GPU.

The obtained classification results, depending on the classifier used for separate types of tennis movements, are presented in Table 1 and Table 2. The number of epochs for training sets for the ST-GCN and the Fuzzy ST-GCN is presented in Table 3.

## 4. Discussion

It has been stated that for 3D motion data the satisfactory results for activity recognition are obtained using a Spatial-Temporal Graph Convolutional Network [42,43]. In this paper, this network was implemented to recognise tennis forehand and backhand shots based on images that represent a model of an athlete together with a tennis racket. An input graph was created using three motion data images of one type of shots. The analysis consists of two approaches: one applying fuzzy input to the graph (Fuzzy ST-GCN) and one without it (ST-GCN). A better accuracy of tennis shot recognition of the two studies described was reached for the Fuzzy ST-GCN (Figure 4 and Figure 5). The use of the Fuzzy ST-GCN ensures quite good recognition results, which confirms the belief that the use of fuzzy sets significantly increases the quality of classification. Based on the analysis of the results for the ST-GCN (Figure 4), a 74.5–75.9% accuracy level of tennis shot recognition was obtained with a training set ranging 40% to 65%. While using the Fuzzy ST-GCN, for a training set equal to or higher than 45% the accuracy obtained was not less than 82.2% (Figure 5). More similar results in differences in accuracy of tennis forehand and backhand shots are obtained for the Fuzzy ST-GCN classifier than for the ST-GCN classifier (Table 1 and Table 2). In addition, the number of epochs for the Fuzzy ST-GCN is smaller than for the ST-GCN (Table 3), which means that the Fuzzy ST-GCN learns faster than the ST-GCN. Based on the obtained results the stated thesis that the Fuzzy ST-GCN classifier is more suitable for tennis forehand and backhand recognition than the ST-GCN classifier was proved. The advantage of the database of images generated from the motion capture system is that it has successive player and racket settings during each shot in successive time units. This allows you to create a time series for the analysed impact. This approach is necessary when applying the ST-GCN classifier. The tennis shot database described in [29], containing a single image for one stroke, cannot be used in the discussed classifier. The maximum accuracy for a Fuzzy ST-GCN network obtained in the presented study is higher than in papers considering other classifiers like Incepction [32], SVM [34], SVM and CRF [33], CNN and LSTM [1]. However, these methods were performed on videos, not pictures obtained directly from a motion capture system.

## 5. Conclusions

The new approach presented for tennis shot recognition using an ST-GCN network gives satisfactory results. The obtained data confirm that applying fuzzy input to the graphs increases the recognition accuracy. Further work will focus on extending the base with new types of strokes (voley, smash and slice). The present approach and the applied neural networks will then be an appropriate tool for the analysis and classification of the type of tennis movement.

The next direction of research may be the analysis of tennis hand movements, selected body elements, and finally the analysis of the movements of the whole body. The correctness of the stroke and its effectiveness usually depend to a large extent on the range of the movements performed. The movement patterns of the shots analysed will be determined. The movements of individual elements of the tennis player’s body will be analysed.

## Figures and Tables

**Figure 1 sensors-20-06094-f001:**
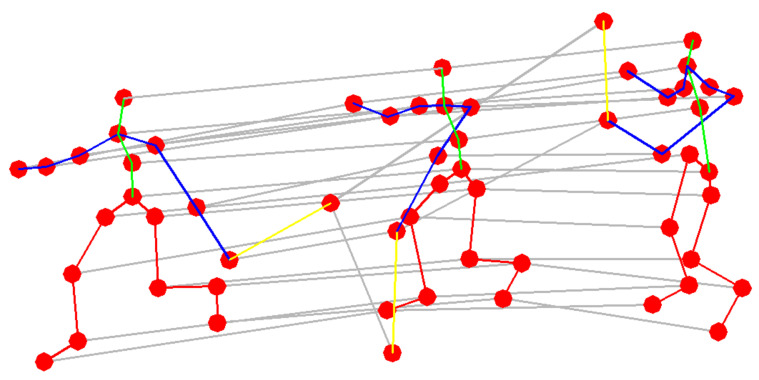
A spatial temporal graph of skeleton. Red dots represent joints and other characteristic points. Red lines represent connections between points within the lower limbs, blue—upper limbs, green—spine, yellow—tennis racket.

**Figure 2 sensors-20-06094-f002:**
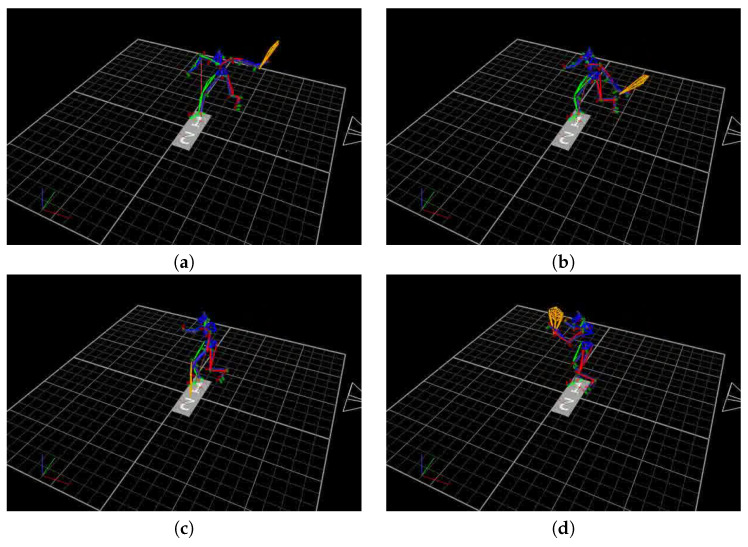
An example of raw forehand shot phases. (**a**) beginning of the preparation phase, (**b**) end of the preparation phase (**c**) hitting the ball and (**d**) swinging the racket after the hit.

**Figure 3 sensors-20-06094-f003:**
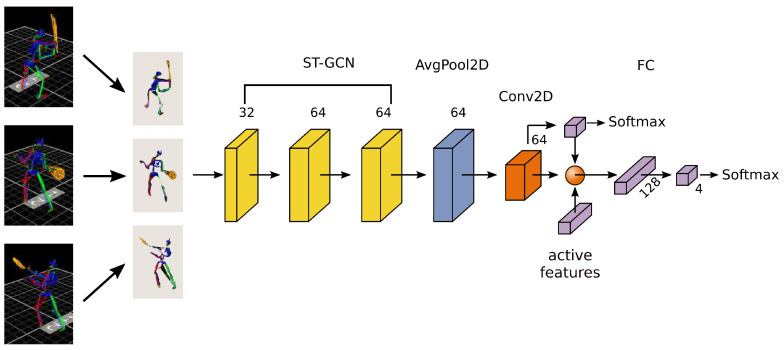
Scheme of used classifier consisting of the ST-GCN part and the active features knowledge base.

**Figure 4 sensors-20-06094-f004:**
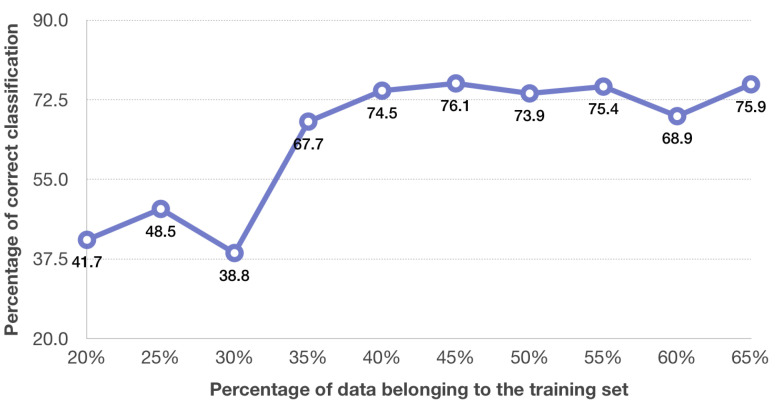
Efficiency plot of ST-GCN classifier.

**Figure 5 sensors-20-06094-f005:**
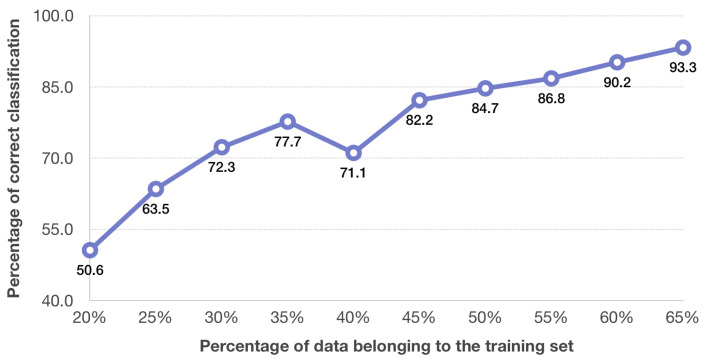
Efficiency of Fuzzy ST-GCN classifier.

**Table 1 sensors-20-06094-t001:** Classification results obtained for the ST-GCN network.

Percentage of Data Belonging to the Training Set	Forehand (True Possitive)	Backhand (True Possitive)	No Shot (True Possitive)
45%	74.8%	74.3%	78.2%
50%	74.4%	74.2%	73.1%
55%	75.6%	76.1%	74.5%
60%	68.5%	74.3%	64.1%
65%	81.2%	77.1%	69.6%

**Table 2 sensors-20-06094-t002:** Classification results obtained for the Fuzzy ST-GCN network.

Percentage of Data Belonging to the Training Set	Forehand (True Possitive)	Backhand (True Possitive)	No Shot (True Possitive)
45%	74.8%	74.3%	78.2%
50%	84.4%	84.1%	78.1%
55%	85.4%	86.2%	82.5%
60%	86.5%	87.3%	86.3%
65%	91.2%	92.8%	95.9%

**Table 3 sensors-20-06094-t003:** The number of epochs.

Percentage of Data Set	ST-GCN	Fuzzy ST-GCN
45%	648	653
50%	525	507
55%	497	482
60%	484	416
65%	473	362

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
