# Peer review of "Learning Three Dimensional Tennis Shots Using Graph Convolutional Networks"

_sensors, 2020, doi:10.3390/s20216094_

Round 1

Reviewer 1 Report

The paper presents a method for recognizing forehand and backhand tennis shots using graph convolutional network. The area is attractive and it could be important for tennis professionals.

The paper is generally well written but there are some comments/recommendations that authors should incorporate in the manuscript:

- Authors state that "According to the authors’ knowledge, there is no research considering the recognition of tennis movements using graphs together with images consisting of models of both the human body and a tennis racket." - There are a plenty of studies which are working on detection and recognition of tennis strokes and related work should also include those studies. And although the results can not be directly compared, they should be discussed/analyzed in the context of other works' performance in the Discussion section.

- "Ten tennis coaches (aged 23.85 ± 5.82, height 1.80 ± 0.11 m, weight 73.21 ± 8.98 kg) took part in the research. Nine were right-handed and one was left-handed." - Ten participants are a small number. How were participants selected? Are only men conducted in the survey? Is there any meaning why nine of them are right handed, and one of them is left handed?

- "In practice, the input feature map is represented by a tensor of (C, V, T) dimensions." - V describes all joints in a skeleton, but C and T should also be clarified.
- Better explanation of each layer in ST-GCN network would be beneficial.
- English should be checked carefully. E.g. line 44: inout --> input, line 68: space is missing before "ST-GCNs", line 178: examplpe --> example, etc.

Reviewer 2 Report

This paper mainly presents a novel approach to recognize the different movements in a tennis game by using graph convolutional network. To verify the impact of input data, two methods are used to train the spatial-temporal graph neural networks. Finally, the experimental results demonstrate the use of fuzzy input graphs for the proposed approach is a better tool for the recognition of forehand and backhand tennis shots. Overall this paper has a logical, intuitive flow and has detailed results. The paper fits well within the scope of the journal. However, I have the following points that should be addressed before it is finally published in the journal.

  • The author presents the related methods for recognizing human movements, but the literature analysis seems insufficient. In order to strengthen control motivation, it is better to enrich the literature review and summarize the main contributions in this point by point.
  • Human movement recognition based on a neural network is a hot research field at present. What are the main challenges for human movement recognition? In this article, how did you solve these problems? It is better to give more analysis of your highlights in this paper.
  • Before discussing the experimental results, it is recommended to give the details of experimental procedures, such as the hardware platform used to train the model.
  • The conclusion part can be expanded. In order to let readers better understand future work, please give specific research directions. It is suggested to read the following manuscripts: Improved recurrent neural network-based manipulator control with remote center of motion constraints: Experimental results; A Smartphone-based Adaptive Recognition and Real-time Monitoring System for Human Activities; Deep Neural Network Approach in Robot Tool Dynamics Identification for Bilateral Teleoperation; Improved human–robot collaborative control of redundant robot for teleoperated minimally invasive surgery
  • Language expression needs further improvement to make the article more readable. It is recommended to double-check the spelling of words and sentence grammar. For example,“the recognition of forehand and backhand tennis shots relative to graphs without fuzzy inout”in the second page. Please check the overall paper carefully. And revise the expression seriously.
